# Melatonin Inhibits Apoptosis and Oxidative Stress of Mouse Leydig Cells via a SIRT1-Dependent Mechanism

**DOI:** 10.3390/molecules24173084

**Published:** 2019-08-25

**Authors:** Gaoqing Xu, Jing Zhao, Hongyu Liu, Jun Wang, Wenfa Lu

**Affiliations:** College of Animal Science and Technology, Jilin Agricultural University, Changchun 130118, China

**Keywords:** melatonin, apoptosis, oxidative stress, SIRT1, Leydig cells, mouse

## Abstract

The purpose of the present study is to examine the effects of melatonin on apoptosis and oxidative stress in mouse Leydig cells and to elucidate the mechanisms responsible for these effects. Our results indicated that 10 ng/mL of melatonin significantly promoted cell viability, the ratio of EdU-positive (5-Ethynyl-2′-deoxyuridine) cells, and increased the mRNA expression of proliferating cell nuclear antigen (*PCNA*), cyclin D1(*CCND1*), and cell division control protein 42 (*CDC42*) (*p* < 0.05). We also observed that melatonin inhibited apoptosis of mouse Leydig cells, accompanied with increased B-cell lymphoma-2 (*BCL-2*) and decreased BCL2 associated X (*BAX*) mRNA and protein expression. Moreover, addition of melatonin significantly decreased the reactive oxygen species (ROS) production and malondialdehyde (MDA) and 8-hydroxy-2′-deoxyguanosine (8-OHdG) levels, while it increased superoxide dismutase (SOD) and glutathione peroxidase (GSH-Px) levels (*p* < 0.05). In addition, we also found that melatonin increased the expression of *SIRT1* (Silent information regulator 1) (*p* < 0.05). To explore the role of SIRT1 signaling in melatonin-induced cells, mouse Leydig cells were pretreated with EX527, an inhibitor of SIRT1. The protective effects of melatonin on mouse Leydig cells were reversed by EX527, as shown by decreased cell proliferation and increased cell apoptosis and oxidative stress. In summary, our results demonstrated that melatonin inhibited apoptosis and oxidative stress of mouse Leydig cells through a SIRT1-dependent mechanism.

## 1. Introduction

Infertility is a medical problem worldwide and approximately half of all cases are caused by male infertility [1]. Several studies have suggested that oxidative stress and apoptosis are important causes of male infertility [2,3,4]. Leydig cells, which are one of the main cell types in the testis and the primary site for androgen synthesis and secretions, can promote the development of the reproductive organ and spermatogenes in males. Additionally, oxidative stress and apoptosis in Leydig cells are closely related to testicular function [5,6]. Therefore, anti-oxidation and anti-apoptosis supplements are a potential strategy to overcome male infertility.

Melatonin (*N*-acetyl-5-methoxy tryptamine) related to both nervous stimulation and endocrine secretion, is mainly produced by the pineal gland, and is also beneficial to the regulation of circadian rhythms [7]. Melatonin is essential to various biological activities, including anti-aging [8], anti-oxidation [9], anti-inflammation [10], anti-apoptosis [11] and anti-pyroptosis [12]. Previous studies have suggested that melatonin could decrease the levels of apoptosis and oxidative stress in a variety of cells, such as granulosa cells [11,13], skeletal muscle cells [14], nucleus pulposus cells [15], and spermatogonial stem cells [16]. In addition, melatonin can directly act on the testis and ameliorate testicular damages caused by oxidative stress, apoptosis, and inflammation [17,18,19,20]. Significantly, past research has depicted that melatonin reduces testosterone secretion in mouse Leydig cells [21,22]. However, the effects of melatonin on apoptosis and oxidative stress in mouse Leydig cells and its mechanisms have not been examined.

Silent information regulator 1 (SIRT1), a preserved nicotinamide adenine dinucleotide (NAD+) dependent histone deacetylase, plays a crucial function in a variety of molecular mechanisms. Some studies have shown that SIRT1 has protective effects against apoptosis and oxidative stress. For example, activating SIRT1could inhibit the apoptosis induced by hypoxia/reoxygenation in H9c2 cells [23]. Additionally, SIRT1 can fight against oxidative stress by protecting osteocytes and bone precursor cells in order to restitute a normal bone remodeling activity in osteoporosis [24]. SIRT1 deacetylation can activate forkhead box O1 (FoxO1) to synthesize superoxide dismutase (SOD) and Catalase (CAT), resulting in increased cellular resistance to oxidative stress [25]. In addition, melatonin has been a potent regulator of SIRT1 and can regulate apoptosis, oxidative stress, inflammation, and autophagy through the SIRT1 signaling pathway, which is found in cellular or animal models of many diseases [10,26,27,28,29]. There is little research on the role of SIRT1 in the absence of pathological conditions. Based on the aforementioned studies, we hypothesized that SIRT1 is engaged in melatonin-regulated apoptosis and oxidative stress in mouse Leydig cells.

Hence, the existing study was conducted to evaluate the protective effects of melatonin against apoptosis and oxidative stress in mouse Leydig cells and the function of SIRT1 signaling in this process.

## 2. Results

### 2.1. Melatonin Promoted Proliferation of Mouse Leydig Cells

First, an MTT assay was used to examine the effect of melatonin on cell viability of mouse Leydig cells. As depicted in Figure 1A, cell viability of Leydig cells treated with 10 and 100 ng/mL melatonin at 36 h or 48 h was enhanced significantly (*p* < 0.05) and the effect at 36 h was better than that at 48 h. However, there was no significant difference among the five groups at 24 h (*p* > 0.05). Therefore, treatment with melatonin for 36 h was selected for the following experiment. Next, we tested the mRNA expression of proliferation related genes, including proliferating cell nuclear antigen (PCNA), cyclin D1 (CCND1), and cell division control protein 42 (CDC42). As shown in Figure 1B–D, 10 ng/mL of melatonin significantly increased the ratio of 5-ethynyl-2′-deoxyuridine (EdU)-positive cells and the mRNA expression of PCNA, CCND1, and CDC42 (*p* < 0.05). These results showed that melatonin promoted proliferation of mouse Leydig cells.

### 2.2. Melatonin Inhibited Apoptosis of Mouse Leydig Cells

We further examined the regulation of melatonin on apoptosis of mouse Leydig cells. First, the apoptosis rate of mouse Leydig cells treated with varying doses of melatonin for 36 h was detected by flow cytometry analysis. Melatonin at concentrations of 10 and 100 ng/mL significantly decreased the apoptosis rate of mouse Leydig cells (Figure 2A) (*p* < 0.05). In addition, when compared with the control group, 10 ng/mL of melatonin significantly decreased the mRNA and protein expression of BCL2 associated X (BAX), while it enhanced the mRNA and protein expression of B-cell lymphoma-2 (BCL-2) (Figure 2B–D) (*p* < 0.01). Together, these data suggested that melatonin inhibited apoptosis of mouse Leydig cells.

### 2.3. Melatonin Suppressed Oxidative Stress of Mouse Leydig Cells

To examine the effect of melatonin on the oxidative stress of mouse Leydig cells, we detected the levels of reaction oxygen species (ROS), malondialdehyde (MDA), 8-hydroxy-2′-deoxyguanosine (8-OhdG), superoxide dismutase (SOD), and glutathione peroxidase (GSH-Px) in mouse Leydig cells after treatment with various concentrations of melatonin. The results of flow cytometry indicated that melatonin at concentrations of 1, 10, 100, and 1000 ng/mL significantly reduced the fluorescence intensity of ROS (*p* < 0.05) and 10 ng/mL of melatonin was the best among the three concentrations (Figure 3A). Additionally, MDA (Figure 3B) and 8-OHdG (Figure 3C) levels (*p* < 0.01) decreased significantly, while SOD (Figure 3D) and GSH-Px (Figure 3E) levels increased (*p* < 0.01). These results showed that melatonin inhibited oxidative stress in mouse Leydig cells.

### 2.4. Melatonin Increased Cell Proliferation via a SIRT1-Dependent Mechanism in Mouse Leydig Cells

Based on the above results, we selected 10 ng/mL of melatonin to treat mouse Leydig cells for 36 h for the next experiment. As depicted in Figure 4A,B, melatonin enhanced the mRNA expression of SIRT1 (*p* < 0.01) and the protein expression of SIRT1 (*p* < 0.05). Subsequently, we found that the addition of EX527 reversed the effects of melatonin on cell viability (Figure 4C) and cell proliferation (Figure 4D,E). EX527 pretreatment also reversed the melatonin-induced decrease in PCNA, CCND1, and CDC42 expression (Figure 4F–H) (*p* < 0.05). There was no significant difference observed between the control group and the EX527-only treatment group (*p* > 0.05). The above results showed that melatonin could promote the proliferation of mouse Leydig cells via a SIRT1-dependent mechanism.

### 2.5. Melatonin Inhibited Apoptosis of Mouse Leydig Cells via a SIRT1-Dependent Mechanism

As shown in Figure 5A, EX527 significantly relieved the apoptosis rate in mouse Leydig cells decreased by melatonin (*p* < 0.01). Additionally, in the melatonin-only treatment, the mRNA and protein expression levels of BCL-2 were enhanced, while BAX expression was decreased. When the cells were pretreated with EX527 followed by melatonin, the mRNA and protein expression levels of BCL-2 and BAX were comparable to those of the control (Figure 5B–D) (*p* < 0.05). There was no significant difference observed between the control group and the EX527-only treatment group (*p* > 0.05). Taken together, melatonin could inhibit apoptosis of mouse Leydig cells via a SIRT1-dependent mechanism.

### 2.6. Melatonin Inhibited Oxidative Stress of Mouse Leydig Cells via a SIRT1-Dependent Mechanism

To examine the role of SIRT1 on oxidative stress inhibited by melatonin in mouse Leydig cells, we detected the levels of ROS, MDA, 8-OHdG, SOD, and GSH-Px in mouse Leydig cells after treatment with melatonin and EX527. The results indicated that, compared to melatonin-only treatment, Ex527 significantly increased the fluorescence intensity of ROS (*p* < 0.05) (Figure 6A). Moreover, Ex527 reversed the decrease in MDA (Figure 6B) and 8-OHdG (Figure 6C) levels (*p* < 0.01) and reversed the increase in SOD (Figure 6D) and GSH-Px (Figure 6E) levels (*p* < 0.01). These results showed that melatonin could inhibit oxidative stress via a SIRT1-dependent mechanism in mouse Leydig cells.

## 3. Discussion

Apoptosis and oxidative damage of Leydig cells severely affect the function of the testes, leading to male infertility [6]. However, the effects of melatonin on apoptosis and oxidative stress in mouse Leydig cells have not yet been studied in detail. Our results indicated that melatonin could inhibit apoptosis and oxidative stress of mouse Leydig cells via the SIRT1 signaling pathway.

Our results showed that melatonin significantly promoted cell proliferation, manifesting as the increases in cell viability, ratio of EdU-positive cells, and related genes expression (PCNA, CCND1, and CDC42). Melatonin also prevented apoptosis of mouse Leydig cells, accompanied with an increase of mRNA and protein expression of BCL-2 and a decrease of BAX expression. In the present study, melatonin sustained BCL-2 expression and reversed bisphenol A-induced apoptosis in the rat testes and epididymal sperm [5]. The anti-apoptotic effect of melatonin has been shown in a variety of cells. For instance, in the bovine ovarian granulosa cells, melatonin not only directly inhibits apoptosis through its receptors MT1 and MT2, but also attenuates apoptosis induced by β-zearalenol and HT-2 toxin [13,30]. Additionally, in mouse cardiomyocytes, melatonin reduced the apoptosis level and the secretion of serum creatine phosphokinase and lactate dehydrogenase, up-regulated the expression of SIRT1 and anti-apoptotic gene BCL-2, and down-regulated the expression of proapoptotic genes BAX and Caspase-3 [31,32]. Similarly, melatonin could increase the mitochondrial membrane potential (MMP) and regulate apoptosis related genes expression, thereby alleviating the AlCl3-induced apoptosis in the rat spleen [33]. All of the aforementioned studies support our results that melatonin has an anti-apoptotic effect on Leydig cells.

Interestingly, melatonin at a concentration of 10 ng/mL showed the best activity, while higher concentrations had no effect. This result was also found in other studies. Treatment of mouse granulosa cells with 0.1, 1, 10, and 100 μM of melatonin revealed that the optimal concentration to inhibit palmitic acid-induced apoptosis was 10 μM, rather than 100 μM [11]. In pig granulosa cells, a low concentration (0.1 μM) of melatonin instead of the high concentration (10 μM) stimulates the synthesis of estradiol and produces differentially expressed genes, which are associated with regulation of cell proliferation, cell cycle, and anti-apoptosis [34]. These results support our findings.

Furthermore, oxidative stress has been closely related to male infertility [1]. Oxidative stress is characterized by increased lipid peroxidation levels and decreased antioxidant enzyme activity, resulting in an imbalance between oxidation and antioxidation [35]. Therefore, increasing ROS content can cause oxidative stress, which leads to DNA damage and apoptosis [4]. MDA is a direct product of lipid peroxidation and 8-OHdG is a common biomarker for DNA oxidative damage. SOD is an important scavenger for ROS and GSH-Px is also one of the indicators of anti-peroxidation ability. A previous study in Leydig cells showed that rutin reduces H2O2-induced oxidative damage by decreasing ROS and MDA levels while increasing GSH, SOD, CAT, and POD activities. This indicates that rutin has strong antioxidant capacity to overcome male infertility [36]. In the present study, we found that melatonin could scavenge ROS and increase SOD and GSH-Px levels, while it decreased MDA and 8-OHdG levels. Therefore, melatonin demonstrated its antioxidative effects in Leydig cells. In addition, melatonin inhibits oxidative AlCl_3_-induced spleen stress in rats by scavenging ROS and decreasing MDA levels, while increasing the activities of SOD and CAT [33]. Similarly, melatonin enhances the levels of the antioxidant enzymes SOD, CAT, and GSH-Px, resulting in attenuation of testicular oxidative stress induced by dexamethasone [37]. Moreover, melatonin relieves DNA damage and oxidative stress of mouse granulosa cells by decreasing the 8-OHdG level, which is consistent with results in HL-60 cells [38,39]. These results strongly suggest that melatonin is an extremely effectual direct free radical scavenger and an antioxidant that acts indirectly in Leydig cells.

SIRT1, a component part of class III histone deacetylase, plays a vital function in apoptosis and oxidative stress. Melatonin is an activator of SIRT1, which was also showed in this study. Nonetheless, the function of SIRT1 in melatonin-regulated apoptosis and oxidative stress in mouse Leydig cells is unclear. Our study demonstrated that EX527, a SIRT1 inhibitor, reversed the preservation effects of melatonin on apoptosis and oxidative stress of mouse Leydig cells. Some studies support this result. The activation of SIRT1, cerebral-protection of melatonin against oxidative stress and apoptosis, relieves cerebral ischemia reperfusion injury, which was reversed by EX527 [40]. Additionally, a previous study suggested that melatonin protected against chronic obstructive pulmonary disease by preventing apoptosis and ER stress through the upregulation of SIRT1 expression in rat bronchial and alveolar epithelial cells [41]. Taken together, our results support the hypothesis that melatonin inhibits apoptosis and oxidative stress via a SIRT1-dependent mechanism in mouse Leydig cells, although the direct targets of SIRT1 in this process and experiments in vivo have not been studied.

## 4. Materials and Methods

### 4.1. Cell Culture and Treatments

The Leydig cell line TM3 was acquired from Procell (CL-0234, Wuhan, China) and chosen as the cellular model in this research study. Cells were cultured in Dulbecco’s modified Eagle medium/F12 (DMEM/F-12, Gibco, Waltham, MA, USA) with 5% super horse serum, 2.5% fetal bovine serum (FBS), and 1% penicillin/streptomycin (Sangon Biotech, Shanghai, China). The cells were incubated in a cell incubator with 5% CO_2_ at 37 °C. Melatonin (M5250-1G, Sigma-Aldrich, St. Louis, MO, USA) was liquified in alcohol at 1 mg/mL and stored at −20 °C. Then, 1 mg/mL of melatonin was diluted in a serum-free medium to final concentrations of 1, 10, 100, and 1000 ng/mL before treatment. In order to avoid the strong stimulation of horse serum, when treated with melatonin, cells were cultured in serum-free medium. To select the appropriate melatonin concentration and incubation time, cells were treated with diverse dosages of melatonin (1, 10, 100, and 1000 ng/mL) at respective time periods (24 h, 48 h, and 72 h). The role of SIRT1 signaling on the effects of melatonin in mouse Leydig cells was explored by pretreating cells for 2 h with the SIRT1 inhibitor EX527 (10 μM, Med Chem Express, NJ, USA).

### 4.2. Cell Viability Assay

The Leydig cells were seeded at a density of 1 × 10^4^ cells in each well of the 96-well plates in triplicate and cultured overnight. After cells were treated with melatonin, 20 μL MTT (5 mg/mL in PBS) were added to each well and then cultured for 4 h at 37 °C with 5% CO_2_. The supernatant was discarded and 160 μL of dimethyl sulfoxide (DMSO) was supplemented to each well. The plates were agitated for 10 min and the absorbance was calculated at a wavelength of 490 nm by a microplate reader (BioTek, Winooski, VT, USA). The expression of cell viability was the magnitude of absorbance values against the control group. This experiment was performed three times independently.

### 4.3. EdU Assay

The effects of melatonin or EX527 on cell proliferation were determined using BeyoClickTM EdU cell proliferation kit with Alexa Fluor 488 (Beyotime Institute of Biotechnology, Shanghai, China). Leydig cells (5 × 10^4^ cells per well) were cultured in triplicate in 24-well plates and treated according to experimental design. Experiments were carried out according to the kit. The proportions of EdU-positive cells were determined by a cell imaging detector (BioTek, Winooski, VT, USA) and calculated based on Gene5 software.

### 4.4. Cell Apoptosis Analysis

After they were collected and washed with PBS, cells were suspended in 1×Binding Buffer at 1 × 10^6^ cells/mL. Next, following the manufacturer’s manual of instruction, 5 μL of Annexin V-FITC and 5 μL of propidium iodide (PI) were added to cell suspension and incubated at room temperature for 15 min. Cell apoptosis was detected and analyzed by flow cytometry analysis (ACEA Biosciences, Hangzhou, China) using a FITC Annexin V Apoptosis Detection Kit I (BD Biosciences, San Jose, CA, USA). Independent experiments were performed three times.

### 4.5. ROS Analysis

The ROS content was measured using ROS Detection Kit following the manufacturer’s instructions (Beyotime Institute of Biotechnology, Shanghai, China). After cells were cultured and treated, cells were collected, washed with PBS, and incubated with DCFH-DA (10 µM) in a cell incubator at 37 °C for 20 min. Then, serum-free cell culture medium was used to wash the cells three times to sufficiently remove the remaining DCFH-DA. The intensity of fluorescence was found using flow cytometry at an excitation wavelength of 488 nm and an emission wavelength of 525 nm. DCF can be detected using the FITC parameter settings because the fluorescence spectrum of DCF is very similar to FITC. Independent scientific research was conducted three times.

### 4.6. Quantitative Reverse Transcription Polymerase Chain Reaction (qRT-PCR) Analysis

Total RNA was isolated from cells using Trizol reagent (Invitrogen, Carlsbad, CA, USA) and then 1 μg was used to combine cDNA by a PrimeScript RT reagent Kit with gDNA Eraser (Takara, Tokyo, Japan). Real-time quantitative PCR was performed using a SYBR^®^ Premix Ex TaqTM II (Takara, Tokyo, Japan) on a Real-Time PCR Detection System (Agilent StrataGene Mx3005P, Santa Clara, CA, USA). The primers sequences of BCL-2, BAX, PCNA, CCND1, CDC42, SIRT1, and β-actin are shown in Table 1. Relative gene expression was calculated with the 2^−∆∆Ct^ method.

### 4.7. Measurement of MDA, 8-OHdG, SOD and GSH-Px

After cell culture and treatment, cell supernatant was collected, centrifuged at 2000 rpm for 20 min at 4 °C, and stored at −20 °C for subsequent analysis. According to the manufacturers’ instructions (Shanghai Langdun Biotech, Shanghai, China), 50 μL of supernatant and 50 μL of biotin-labeled recognition antigen were added to each well in triplicate and incubated at 37 ° C for 30 min. After washing with PBST, avidin-HRP was added and incubated at 37 ° C for 30 min. The chromogen solution and the stop solution were added to each well after washing again, and the levels of MDA, GSH-Px, 8-OHdG were detected at 450 nm using a microplate reader. Additionally, according to the manufacturers’ instructions (Shanghai Jianglai Biotech, Shanghai, China), 10 μL of supernatant, 40 μL of sample diluent, and 100 μL of HRP-conjugate reagent were added to each well and incubated at 37 ° C for 1 h. After washing and chromogen, the SOD level was detected at 450 nm using a microplate reader.

### 4.8. Western Blot Analysis

Total protein was extracted using RIPA lysis buffer (Beyotime Institute of Biotechnology, China) and then measured using the bicinchoninic acid (BCA) Protein Assay kit (Beyotime Institute of Biotechnology, Shanghai, China). The proteins were detached using SDS-PAGE and transferred to NC membranes (Merck Millipore, Darmstadt, Germany). After the membranes were blocked with Odyssey Blocking Buffer (LI-COR Biosciences, USA) for 1.5 h at 37 °C, the membranes were incubated with primary antibodies against BCL-2, BAX, SIRT1, and β-actin (Table 2) at 4 °C overnight. Then, the membranes were washed with TBST and incubated with goat anti-rabbit IgG or goat anti-mouse IgG for 1 h at 25 °C. The protein bands were detected by a chemisope imaging system (CLiNX Science Instruments, Shanghai, China) and the value for the control group was set as 100%.

### 4.9. Statistical Analysis

Data were presented as mean ± standard error of the mean (SEM). Differences between groups were ascertained by one-way analysis of variance (ANOVA) and Student’s t test. The *p*-values < 0.05 were regarded to be statistically significant. All statistical analyses were performed using GraphPad Prism version 5.0 (GraphPad Prism Software, San Diego, CA, USA).

## 5. Conclusions

In conclusion, our studies demonstrated that melatonin promoted proliferation and inhibited apoptosis, while it decreased the level of ROS and oxidative stress in Leydig cells. Moreover, our results showed that melatonin exerted its anti-apoptotic and anti-oxidative effects by activating the SIRT1 signal pathway in mouse Leydig cells. Therefore, our findings provide strong evidence that melatonin has the potential to treat male infertility.

## Figures and Tables

**Figure 1 molecules-24-03084-f001:**
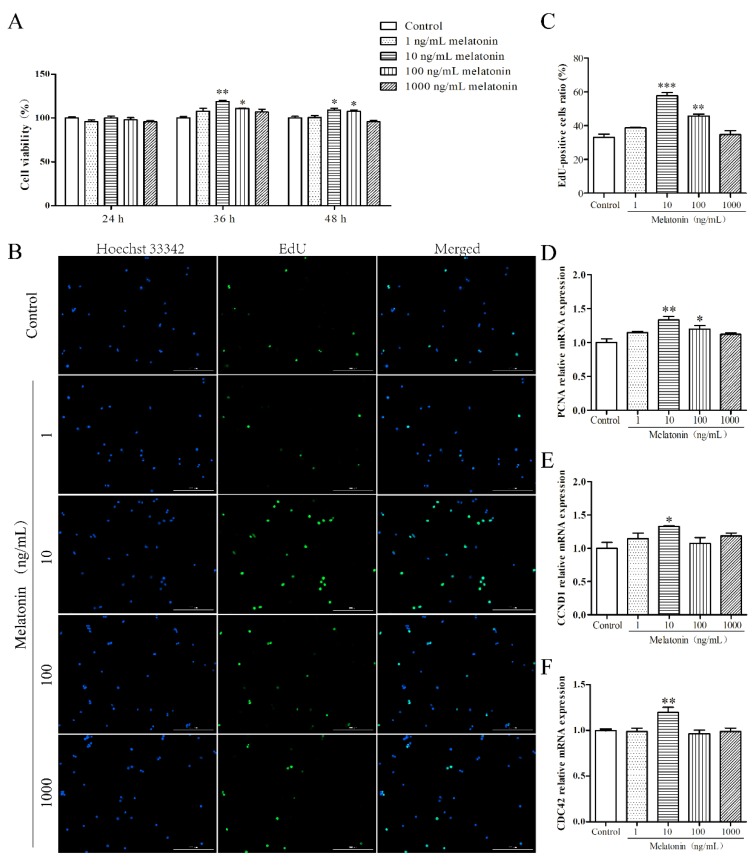
Effects of melatonin on proliferation of mouse Leydig cells. (**A**) The effects of different concentrations (1, 10, 100, and 1000 ng/mL) of melatonin on the cell viability of mouse Leydig cells at various times (24, 48, and 72 h) (*n* = 3). (**B**) Proliferation of mouse Leydig cells treated with different concentrations of melatonin was measured using the EdU incorporation assay (*n* = 3). Green fluorescence represents EdU-labeled Leydig cells (original magnification ×10). (**C**) The proportion of EdU-positive Leydig cells as shown in panel (**B**). The relative mRNA expression levels of proliferating cell nuclear antigen (*PCNA*) (**D**), cyclin D1 (*CCND1*) (**E**), and cell division control protein 42 (*CDC42*) (**F**) (*n* = 3). Values are shown as mean ± SEM. *** *p* < 0.001, ** *p* < 0.01 or * *p* < 0.05 compared with the control group.

**Figure 2 molecules-24-03084-f002:**
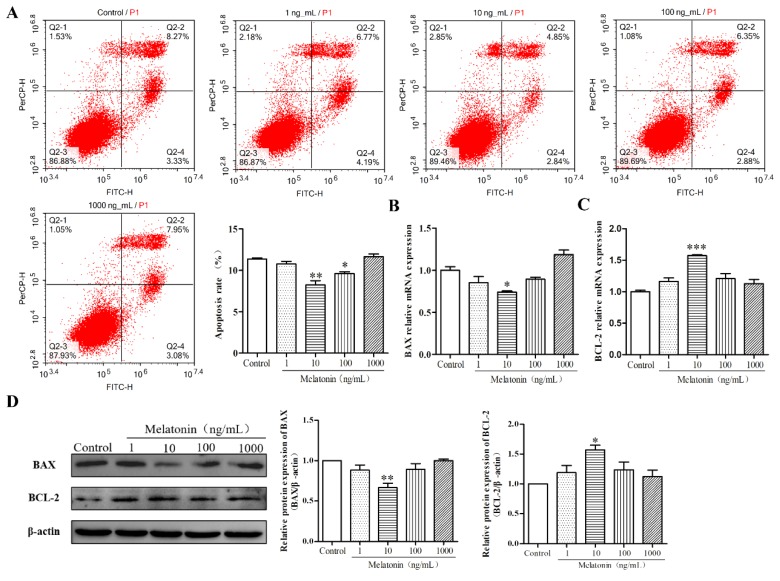
Effects of melatonin on regulating the mRNA and protein expression levels of apoptosis related factor. (**A**) The effects of different concentrations of melatonin on apoptosis rate of mouse Leydig cells for 36 h (*n* = 3). The four quadrants in the figure represent dead cells (Q2-1), late-stage apoptotic cells (Q2-2), viable cells (Q2-3), and early-stage apoptotic cells (Q2-4). The apoptosis rate is the sum of values from Q2-2 and Q2-4. The relative mRNA expression levels of *BAX* (**B**) and *BCL-2* (**C**) (*n* = 3). (**D**) The relative protein expression levels of BAX and BCL-2 were detected and analyzed (*n* = 3). Values are shown as mean ± SEM. *** *p* < 0.001, ** *p* < 0.01 or * *p* < 0.05 compared with the control group.

**Figure 3 molecules-24-03084-f003:**
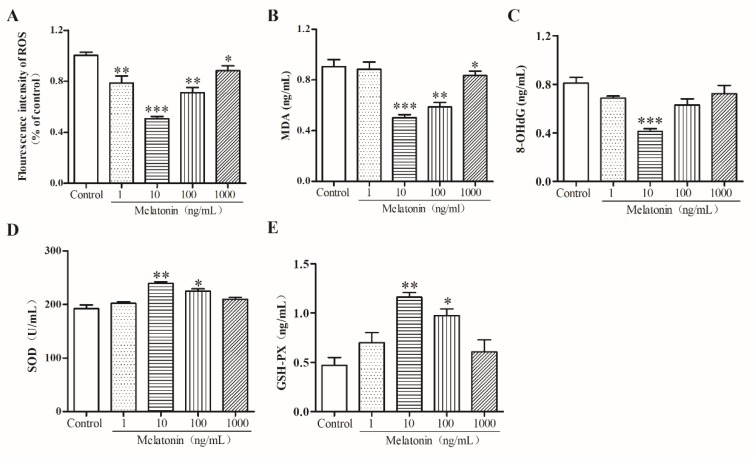
Effects of melatonin on reactive oxygen species (ROS), malondialdehyde (MDA), 8-hydroxy-2′-deoxyguanosine (8-OHdG), superoxide dismutase (SOD), and glutathione peroxidase (GSH-Px) in mouse Leydig cells. (**A**) The fluorescence intensity of ROS was measured by flow cytometry analysis (*n* = 3). The levels of MDA (**B**), 8-OHdG (**C**), SOD (**D**) and GSH-Px (**E**) were measured by a microplate reader (*n* = 3). Values are shown as mean ± SEM. *** *p* < 0.001, ** *p* < 0.01 or * *p* < 0.05 compared with the control group.

**Figure 4 molecules-24-03084-f004:**
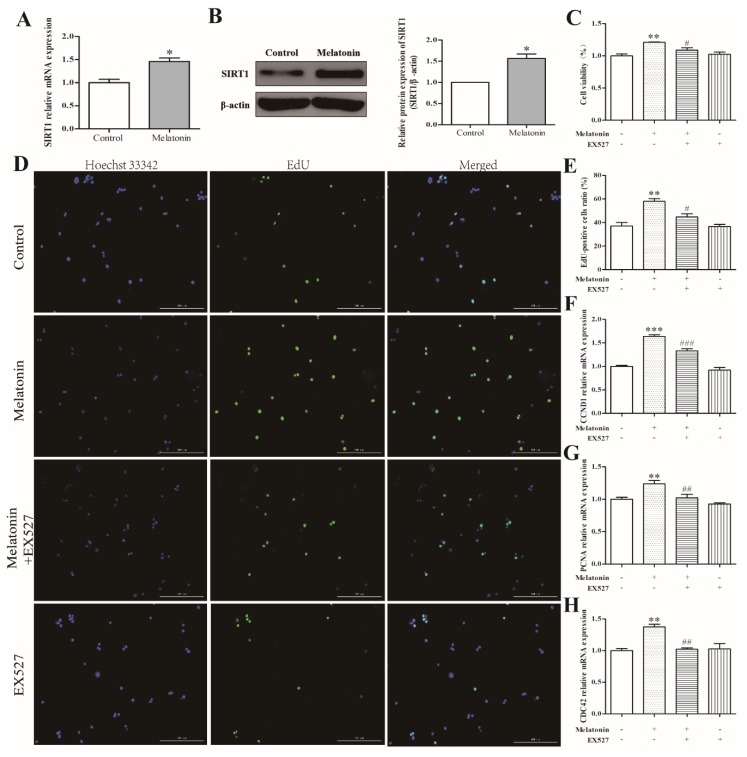
Effects of melatonin and EX527 on cell viability and proliferation related genes of mouse Leydig cells. (**A**) The mRNA expression level of *SIRT1* (*n* = 3). (**B**) The protein expression level of SIRT1 (*n* = 3). (**C**) The effects of melatonin and EX527 on cell viability of mouse Leydig cells (*n* = 3). (**D**) Proliferation of mouse Leydig cells in the control, melatonin, melatonin+EX527, and EX527 groups was measured using the EdU incorporation assay. Green fluorescence represents EdU-labeled Leydig cells (original magnification ×10). (**E**) The proportion of EdU-positive Leydig cells as shown in panel (**D**). The effects of melatonin and EX527 on mRNA expression levels of *CCND1* (**F**), *PCNA* (**G**), and *CDC42* (**H**) in mouse Leydig cells (*n* = 3). Values are shown as mean ± SEM. *** *p* < 0.001, ** *p* < 0.01 or * *p* < 0.05 compared with the control group; ### *p* < 0.001, ## *p* < 0.01 or # *p* < 0.05 compared with the melatonin-only treatment group.

**Figure 5 molecules-24-03084-f005:**
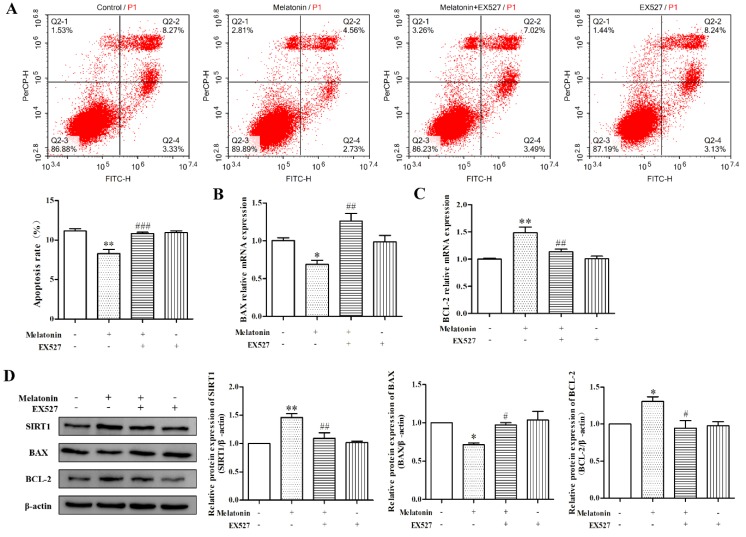
Effects of melatonin and EX527 on apoptosis of mouse Leydig cells. (**A**) The effects of melatonin and EX527 on apoptosis rate of mouse Leydig cells (*n* = 3). The four quadrants in the figure represent dead cells (Q2-1), late-stage apoptotic cells (Q2-2), viable cells (Q2-3), and early-stage apoptotic cells (Q2-4). The apoptosis rate is the sum of values from Q2-2 and Q2-4. The relative mRNA expression levels of *BAX* (**B**) and *BCL-2* (**C**) (*n* = 3). (**D**) The relative protein expression levels of SIRT1, BAX, and BCL-2 were detected and analyzed (*n* = 3). Values are shown as mean ± SEM. ** *p* < 0.01 or * *p* < 0.05 compared with the control group; ### *p* < 0.001, ## *p* < 0.01 or # *p* < 0.05 compared with melatonin-only treatment group.

**Figure 6 molecules-24-03084-f006:**
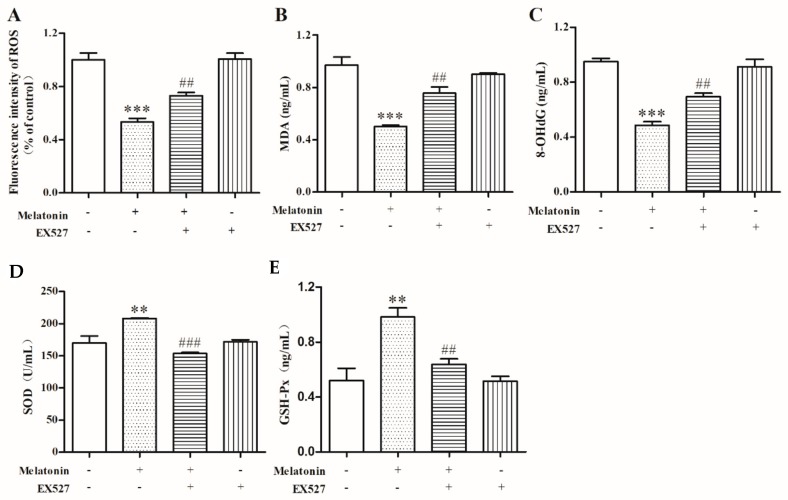
Effects of melatonin and EX527 on ROS, MDA, 8-OHdG, SOD, and GSH-Px in mouse Leydig cells. (**A**) The fluorescence intensity of ROS was measured by flow cytometry analysis (*n* = 3). The levels of MDA (**B**), 8-OHdG (**C**), SOD (**D**), and GSH-Px (**E**) were measured by a microplate reader (*n* = 3). Values are shown as mean ± SEM. *** *p* < 0.001 or ** *p* < 0.01 compared with the control group; ### *p* < 0.001 or ## *p* < 0.01 compared with melatonin-only treatment group.

**Table 1 molecules-24-03084-t001:** Information on primers of real-time quantitative PCR.

Genes	Primer Sequence (5′–3′)	Genebank No.	Size (bp)
BCL-2	F: ACGGTGGTGGAGGAACTCTTCAG	XM_021173243.1	168
	R: GGTGTGCAGATGCCGGTTCAG		
BAX	F: CGTGAGCGGCTGCTTGTCTG	XM_021195914.1	128
	R: ATGGTGAGCGAGGCGGTGAG		
PCNA	F: TGAAGAAGGTGCTGGAGGCTCTC	NM_011045.2	115
	R: AGCTGTACCAAGGAGACGTGAGAC		
CCND1	F: TGGATGCTGGAGGTCTGTGAGG	XM_011241977.1	112
	R: GCAGGCGGCTCTTCTTCAAGG		
CDC42	F: GGCTGTCAAGTATGTGGAGTGCTC	XM_021159845.1	111
	R: CTGCGGCTCTTCTTCGGTTCTG		
SIRT1	F: CGTCTTGTCCTCTAGTTCCTGTG	NM_001159589.2	134
	R: GCCTCTCCGTATCATCTTCCAAG		
β-actin	F: GTGCTATGTTGCTCTAGACTTCG	NM_007393.5	174
	R: ATGCCACAGGATTCCATACC		

**Table 2 molecules-24-03084-t002:** Details of primary and secondary antibodies used for western blot in this study.

Antibodies	Cat NO.	Source	Dilution
BCL-2	A11025	ABclonal, Wuhan, China	1:1000
BAX	A12009	ABclonal, Wuhan, China	1:1000
SIRT1	AB110304	Abcam, Cambridge, UK	1:1000
β-actin	60008-1-lg	ProteinTech, Chicago, IL, USA	1:5000
Goat Anti-Rabbit IgG	SA00001-2	ProteinTech, Chicago, IL, USA	1:5000
Goat Anti-mouse IgG	SA00001-1	ProteinTech, Chicago, IL, USA	1:5000

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
