# Peer review of "Melatonin Inhibits Apoptosis and Oxidative Stress of Mouse Leydig Cells via a SIRT1-Dependent Mechanism"

_molecules, 2019, doi:10.3390/molecules24173084_

Round 1

Reviewer 1 Report

The authors reported that melatonin inhibited apoptosis and oxidative stress 23 of mouse Leydig cell through a SIRT1-dependent mechanism. The research topic is interesting. Some major concernings should be addressed before further consideration. 1) The manuscript was poorly presented. The English must be polished by professional language editor. 2) The proliferative effect of melatonin on Leydig cells could not be determined by only MTT assay. Other assays must be performed. 3) Why only 10 ng/ml melatonin showed activity, but not higher concentrations? 4) SIRT1 is a deacetylase. Could the authors speculate the direct targets of SIRT1 in mediating the effects of melatonin?

Author Response

Dear Reviewer 1:

Thank you for you comments concerning our manuscript entitled “Melatonin inhibits apoptosis and oxidative stress of mouse Leydig cells via a SIRT1-dependent mechanism”. Those comments are all valuable and very helpful for revising and improving our paper, as well as the important guiding significance to our researches. We have studied comments carefully and have made correction and supplement which we hope meet with approval. Revised portion are marked with blue (revised according to reviewers) in the paper. The main corrections in the paper and the response to you comments are as following:

The authors reported that melatonin inhibited apoptosis and oxidative stress of mouse Leydig cell through a SIRT1-dependent mechanism. The research topic is interesting. Some major concernings should be addressed before further consideration.

1) The manuscript was poorly presented. The English must be polished by professional language editor.

Response: Thank you for your suggestion. We have completed the English revision of the manuscript by professional editors at “Editage”, a division of Cactus Communications (Job code: DHGIV_7). Certificate of English editing is among the files we uploaded.

2) The proliferative effect of melatonin on Leydig cells could not be determined by only MTT assay. Other assays must be performed.

Response: This comment is valuable and very helpful for improving our paper. We added a proliferation experiment–EdU (5-Ethynyl-2’-deoxyuridine) assay to demonstrated that melatonin promoted proliferation of mouse Leydig cells by detecting the ratio of EdU-positive cells. We also found that EX527 reversed the proliferative effect of melatonin in the EdU assay.

3) Why only 10 ng/ml melatonin showed activity, but not higher concentrations?

Response: For this problem, we considered that there are two reasons. First, we did not use drug stimulation (e.g. H2O2) to induce apoptosis and oxidative stress in this study, so lower concentrations of melatonin showed better results. Second, melatonin is often used in an effective concentration range, and higher concentrations of melatonin are not effective, which has been reported in many studies. We added discussion of this problem in the article as followed.

Treatment of mouse granulosa cells with 0.1, 1, 10, and 100 μM melatonin revealed that the optimal concentration to inhibit palmitic acid-induced apoptosis was 10 μM rather than 100 μM (Chen Z, 2019). In pig granulosa cells, melatonin with low concentration (0.1 μM) instead of high concentration (10 μM) could stimulate the synthesis of estradiol and make differentially expressed genes which associated with regulation of cell proliferation, cell cycle, and anti-apoptosis significantly enriched (Liu Y, 2019).

4) SIRT1 is a deacetylase. Could the authors speculate the direct targets of SIRT1 in mediating the effects of melatonin?

Response: SIRT1 as an NAD-dependent III histone deacetylase can interact with a variety of target proteins such as FOXO, tumor suppressor p53, and nuclear transcription factor NF-κB. At present, the direct targets of SIRT1 in mediating the effects of melatonin has not been reported. We speculated that SIRT1 could interact with Nrf2 to regulate oxidative stress in this study. However, we did not study the direct targets of SIRT1 in mediating the effects of melatonin, which is a limitation of this article.

Thank you for your review.

Yours Sincerely

Reviewer 2 Report

The manuscript titled "Melatonin inhibits apoptosis and oxidative stress 3 of mouse Leydig cells via a SIRT1-dependent 4 mechanism" by Xu and colleagues is a well organized investigation that may have implications for the etiology and treatment of male infertility. Although well put together, the manuscript would benefit greatly from close English language editing to improve readability and grammar. That aside, I have a few suggestions for improvement:

MTT assay, Figure 1A and/or Materials and Methods: please indicate how statistics were determined, e.g., biological replicates versus replicate wells

Line 80 indicates reductions in apoptosis to 70.2% and 82.3% in reference to Figure 2A. It is not intuitively clear from these images how these values were determined. Clarification would be helpful in the text, emphasizing that the reductions were 70.2% and 82.3% of the apoptotic rate of untreated cells. Also, for those unfamiliar with flow data, indicating which quandrants are considered apoptotic cells would be helpful. In fact, the flow plots might be better placed as a supplemental figure and leave the histogram of the apoptotic rate % as Figure 2A. Just a suggestion for simplification.

Lines 102-103: "three" concentrations should be "five"

Results section 2.6: should be referencing Figure 6 instead of 3 & 5

Lines 146-147: these results indicate only partial reversal of oxidative stress when pretreating with the SIRT1 inhibitor. Therefore, one can only state that inhibition of oxidative stress in mouse Leydig cells is at least partially due to a SIRT1-dependent mechanism, although other mechanisms may also contribute.

Finally, I have a bit of concern regarding the nature of the response and what appears to be a narrow therapeutic window in terms of dose and time. Can the authors speculate on why this is and how it might be relevant in an in vivo setting? This might be something to add to the Discussion.

Author Response

Dear Reviewer 2:

Thank you for your comments concerning our manuscript entitled “Melatonin inhibits apoptosis and oxidative stress of mouse Leydig cells via a SIRT1-dependent mechanism”. Those comments are all valuable and very helpful for revising and improving our paper, as well as the important guiding significance to our researches. We have studied comments carefully and have made correction and supplement which we hope meet with approval. Revised portion are marked with blue (revised according to reviewers) in the paper. The main corrections in the paper and the response to you comments are as following:

The manuscript titled "Melatonin inhibits apoptosis and oxidative stress of mouse Leydig cells via a SIRT1-dependent mechanism" by Xu and colleagues is a well organized investigation that may have implications for the etiology and treatment of male infertility. Although well put together, the manuscript would benefit greatly from close English language editing to improve readability and grammar. That aside, I have a few suggestions for improvement:

1) MTT assay, Figure 1A and/or Materials and Methods: please indicate how statistics were determined, e.g., biological replicates versus replicate wells.

Response: For the question of biological replicates and replicate wells, we explained in materials and methods. The wells are repeated in triplicate and this experiment was performed three times independently. Thank you for your suggestion.

2) Line 80 indicates reductions in apoptosis to 70.2% and 82.3% in reference to Figure 2A. It is not intuitively clear from these images how these values were determined. Clarification would be helpful in the text, emphasizing that the reductions were 70.2% and 82.3% of the apoptotic rate of untreated cells. Also, for those unfamiliar with flow data, indicating which quadrants are considered apoptotic cells would be helpful. In fact, the flow plots might be better placed as a supplemental figure and leave the histogram of the apoptotic rate % as Figure 2A. Just a suggestion for simplification.

Response: Thank you for your suggestion. The suggestion made us realize that our results analysis are not intuitive. These values (70.2% and 82.3%) really can't be seen intuitively in the results. Therefore, we replaced these values with intuitive expressions (significant decrease and P values). In addition, we explained the four quadrants and point out the quadrant of apoptosis in the revision.

3) Lines 102-103: "three" concentrations should be "five"

Response: Thank you very much for your correction. I have modified in the article.

Results section 2.6: should be referencing Figure 6 instead of 3 & 5

Response: This is the negligence in our writing. Thank you very much for your correction. We replaced Figure 3 & 5 with Figure 6 in result section 2.6.

4) Lines 146-147: these results indicate only partial reversal of oxidative stress when pretreating with the SIRT1 inhibitor. Therefore, one can only state that inhibition of oxidative stress in mouse Leydig cells is at least partially due to a SIRT1-dependent mechanism, although other mechanisms may also contribute.

Response: This comment is valuable and very helpful for improving our paper. We realized that this statement is not rigorous and we corrected it.

5) Finally, I have a bit of concern regarding the nature of the response and what appears to be a narrow therapeutic window in terms of dose and time. Can the authors speculate on why this is and how it might be relevant in an in vivo setting? This might be something to add to the Discussion.

Response: The production of free radicals can cause damage to biological macromolecules such as lipids, proteins, and nucleic acids, leading to cell damage. Melatonin can scavenge free radicals, protect against oxidation and inhibit lipid peroxidation to protect cell structures, prevent DNA damage, and reduce peroxide levels in the body.

For the problem of dose and time, we considered that there are two reasons. First, dose and time of melatonin depend on the degree of oxidative stress and apoptosis. If adding drug stimulation, we speculate that melatonin will play a protective role in advance and the concentration will increase. For example, melatonin treatment for 24 h relieves high glucose-induced apoptosis in Schwann cells (Tiong YL, 2019). Second, melatonin is often used in an effective range of concentration dose and time. The dose and time of melatonin required by different individuals is also different, and the individual differences are large. It is necessary to try to determine the appropriate dose. Lower and higher concentrations of melatonin are not effective, which has been reported in many studies. For example, treatment of mouse granulosa cells with 0.1, 1, 10, and 100 μM melatonin revealed that the optimal concentration to inhibit palmitic acid-induced apoptosis was 10 μM rather than 0.1, 1 and 100 μM (Chen Z, 2019). In pig granulosa cells, melatonin with low concentration (0.1 μM) instead of high concentration (10 μM) could stimulate the synthesis of estradiol and make differentially expressed genes which associated with regulation of cell proliferation, cell cycle, and anti-apoptosis significantly enriched. We added discussion of this problem in the article (Liu Y, 2019).

For how it might be relevant in an in vivo setting, the difference between in vivo and in vitro is very large. In vivo experiment, melatonin often requires multiple injections, such as one week or longer. Studies have shown that melatonin injection for 7 days could inhibit apoptosis and oxidative stress of kidney and testis in mice. Melatonin injection for one month treatment may prevent formaldehyde-induced oxidative damage and apoptosis in rat testes. However, we did not study the effects in animals, which is a limitation of thi s article.

Thank you for your review.

Yours Sincerely

Round 2

Reviewer 1 Report

Most of my concerning have been addressed.

Two minor issues. 1) the reference style should be followed the MDPI format and kept consistently. 2) In Figure 3-5, use different statistical markers, such as # EX527 vs vehicle, and * melatonin vs vehicle.

Author Response

Response to Reviewer 1 Comments

Thank you for your comments concerning our manuscript entitled “Melatonin inhibits apoptosis and oxidative stress of mouse Leydig cells via a SIRT1-dependent mechanism”. The two small problems you pointed out are very helpful and valuable to us. We made correction which we hope meet with approval. The response to your comments are as following:

Point 1: the reference style should be followed the MDPI format and kept consistently.

Response 1: We used endnote to insert the reference before, but found some differences in the latest articles. We have modified all reference styles according to the MDPI format.

Point 2: In Figure 3-5, use different statistical markers, such as # EX527 vs vehicle, and * melatonin vs vehicle.

Response 2: Thank you very much for your suggestion. We used different statistical markers (* and #) in the column chart. We modified Figure 4-6 and their legends.